# Creating the Current and Riding the Wave: Persistence and Change in Community-Engaged Health Sciences Research

Krista A. Haapanen [1,*], Jonathan K. London [2] and Karen Andrade [2]

1   Department of Human and Organizational Development, Vanderbilt University, Nashville, TN 37235, USA
2   Department of Human Ecology, University of California, Davis, CA 95616, USA;
    jklondon@ucdavis.edu (J.K.L.); kandradec@gmail.com (K.A.)
*   Correspondence: krista.a.haapanen@vanderbilt.edu

**Abstract:** Recent decades have seen considerable increases in funding and support for community-engaged research (CER) in the health sciences, including the introduction of community engagement requirements into federally funded research infrastructure programs. This paper asks why, despite these supports and incentives, even the best-intentioned researchers and research organizations may struggle to design, implement, and sustain successful community engagement strategies. This question is examined using an exploratory case study of an environmental health sciences research center whose strategies were influenced in part by a requirement by the funder to incorporate community engagement into its research activities. This study utilizes multiple sources of qualitative data collected between the research center's second and fifth years of operation, including participant observation, interviews, and focus groups. The analysis employs an organizational perspective, yielding insights into the factors hindering and facilitating the development of practices that integrate community perspectives and control into academic structures. The findings point to an ongoing dialectic between support for innovative community engagement practices and persistence of conventional academic structures. We highlight the interconnected effects of environmental influences, organizational structures, and individual agency on the development of innovative community engagement practices. The implications for future research and practice are discussed.

**Keywords:** community engagement; community–university partnership; health research; CER; CEnR; CBPR

## 1. Introduction

Community-engaged research (CER) approaches have gained increasing recognition from researchers and funders due to their benefits for the rigor and social impact of health research. These benefits have led the National Institutes of Health (NIH) to build CER into such research infrastructure programs as the Clinical and Translational Science Awards (CTSAs), National Institute of Environmental Health Sciences (NIEHS) Environmental Health Sciences Core (P30) program, and NIEHS Superfund Research Multi-Project (P42) Centers. Grantees of these programs are expected—and in some programs, required—to establish bi-directional communication with their audiences and facilitate the translation of research into relevant public health domains. With over 50 Clinical and Translational Science Awards (NCATS 2023), 26 NIEHS Environmental Health Sciences (P30) Centers (NIEHS 2023a), and 25 NIEHS Superfund (P42) Research Centers (NIEHS 2023b) nationwide, these commitments represent a considerable effort to expand the adoption of CER in the health sciences.

Although such advances support the expansion of CER in health sciences research, studies point to ongoing challenges with the development and implementation of CER programming. Reports from CTSAs, for instance, highlight such obstacles as lack of leadership from the institution and funder, need for capacity development among institutional staff

and community partners, community partners' limited time and funding, and limits on staff time (Holzer and Kass 2015). They suggest that failure to "get" how CER functions differently from traditional biomedical research or to recognize its potential benefits can hinder institutional leaders from providing adequate support or can result in stated support not being borne out in action (Holzer and Kass 2015). For community representatives, this failure of university partners to understand CER may be perceived as a lack of respect for the value of their participation in research (Freeman et al. 2014). Community representatives report being un- or undercompensated, unsure of how their input is ultimately used by the CTSA, treated inequitably in the distribution of funding and in governance/leadership roles, and involved with the CTSA only via the CEC staff (Freeman et al. 2014; Wilkins et al. 2013). These challenges suggest that, despite pressure and support from funders, academic and non-academic entities are failing to develop effective collaborative practices and shared understandings around CER.

Much is already known about the ongoing tensions between the structures of many academic institutions and processes of CER. Academic promotion guidelines and incentives, for instance, were not developed with CER in mind. These systems rarely support or reward the time- and resource-intensive processes of developing relationships with non-academic partners, designing research that aligns with community needs and priorities, and supporting translation of and action on the findings (Wilmsen and Krishnaswamy 2008; London et al. 2020). When researchers and community partners enter the collaboration with differences in their cultural experiences, professional backgrounds, and organizational and personal priorities, tensions can emerge between partners (London et al. 2018). How these tensions are navigated can determine whether CER succeeds in co-creating new knowledge or reinforces academic legacies of ivory tower elitism (Wallerstein et al. 2019; Roura 2021).

However, less scholarship focuses on systemic ways that academic structures may resist the integration of CER practices. Studies that do focus on community-engaged research centers (such as the CTSA program) have mainly looked for program-wide trends in the processes and outcomes of CER (e.g., Freeman et al. 2014; Wilkins et al. 2013). Particularly in the health sciences, studies of academic community engagement are predominantly descriptive and atheoretical, revealing little about the complex and locally situated ways that external norms and messages, internal organizational structures, and actions of individual people coalesce to either facilitate or hinder effective CER. Drawing from organizational theory, we ask how members of community-engaged research centers navigate external pressures to develop effective CER programs and practices.

### 1.1. Institutionalism and the Community-Engaged Health Sciences Research Center

Institutional theory, an area of scholarship that focuses on organizations' relationships with their environments (DiMaggio and Powell 1983), can help to understand research centers' responses to funding environments, professional standards, and the organizational behavior of their peers. In contrast to the theories of interpersonal and group dynamics that have often been used to study CER teams (e.g., Israel et al. 2020), institutional theory frames community-engaged research centers as organizations that must appear legitimate to other organizations in order to secure and maintain funding. DiMaggio and Powell (1983) posited that, in order to survive, organizations will adopt behaviors that signal compliance with external requirements, align with professional standards, and reflect best practices developed and adopted by their peers.

An organizational *field* emerges as "actors with dependent interests and worldviews are . . . forced increasingly to take one another into account in their actions" (Fligstein and McAdam 2012, p. 87). In the case of CER, growing support for community involvement in health and biomedical sciences research (Ortiz et al. 2020; Wilkins and Alberti 2019) has led previously unrelated organizations to align their resources toward common objectives. Community organizations and academic researchers now perceive one another as potential collaborators and allies in efforts to identify, understand, and address health issues in local communities. Institutional theory holds that, as these organizations involved in CER

increasingly interact, respond to similar pressures, and participate in similar activities, they will develop similar behaviors which, over time, crystallize into institutional logics or cultural rules, shared norms, and taken-for-granted understandings of what is appropriate and normal for organizations of that type (DiMaggio and Powell 1983). These social "rules of the game" (Jepperson 1991; Thornton and Ocasio 2008) provide a blueprint for both individual and organizational behavior. Universities, for example, have certain widely recognized structures for administration, roles for faculty and students, and experiences for students such as on-campus living.

In an *emerging* field, these shared understandings have yet to crystallize and are still being negotiated (Fligstein and McAdam 2012). The actions of organizations and individuals in the field reflect differing interpretations of and responses to messages about what structures and behaviors are legitimate (Scott and Davis 2015). Individuals' roles in their organizations, their past experiences and values, and various other factors will shape how they interpret, negotiate, and act upon those messages. We suggest that CER in the health sciences is an emerging field, and that the types of challenges documented by health sciences research centers (e.g., Freeman et al. 2014; Wilkins et al. 2013) indicate that participants in these research centers' activities may not share a set of common understandings about what CER is and should be. Holzer and Kass's (2015) observation that researchers in CTSAs often fail to "get" CER, for instance, points to a lack of consensus around the meaning of and social expectations for CER.

### 1.2. The Current Study

The goal of this exploratory analysis is to generate new insights into the factors that hinder and facilitate academic CER, particularly in the health sciences. We ask why, despite growing support and resources for community-engaged health sciences research in these fields, even the best-intentioned researchers and research organizations may struggle to design and implement sustainable strategies. The article reports on a case study of an Environmental Health Sciences Center (henceforth "the Center") funded by the National Institute of Environmental Health Sciences (NIEHS) Environmental Health Sciences Core Center (P30) program. The authors of this study are: a former program coordinator, a CEC director, and a CEC manager, in order of listing. In accordance with the P30 program's guidelines, the Center was required to incorporate CER into its research activities. We explore the development and implementation of these practices during the Center's first funding cycle. (Note: the Center is finishing its second cycle at the time of this writing and has significantly enhanced its CER practices, based in part on the assessment in this article.).

## 2. Materials and Methods
### 2.1. Setting

The Center was launched in 2015 and began its second five-year funding cycle in April 2020. The research took place between 2016 and 2019. At the time the research was conducted, the Center was one of approximately 20 research centers funded by the National Institute of Environmental Health Sciences (NIEHS) Environmental Health Sciences Core Center (P30) program. The P30 program facilitates scientific collaborations by funding institutional infrastructure to support scientific equipment, facilities, and other resources that can be shared among environmental health researchers (NIEHS 2023a). The Center's work focuses on the most environmentally vulnerable places and populations in California, with the greatest attention to California's Central Valley, a region that faces pressing environmental health concerns such as air pollution, water contamination, pesticide exposure, and climate change-related disasters (Huang and London 2012; London et al. 2011).

The Center provides a relevant case study for a few reasons. First, the P30 program represents a particular commitment to cultivating CER in the environmental health sciences, with all funded centers required to establish and fund a dedicated subunit known as a community engagement core (CEC). The role of the CEC is to communicate environmental health research findings and concepts to community partners and convey the voice of

these communities to researchers within the center (NIEHS 2023a). The Center had also only recently launched when the study began, so its programs—including its strategies for engaging the community—were still being developed, implemented, and negotiated. This provided a useful window into the assumptions and ideologies that the Center's leadership brought to the formation of the Center, the ways that the organization first began to take shape, and the shifts that occurred over time.

In keeping with the P30 program's funding requirements, the Center consisted of seven subunits: community engagement core (CEC), administrative core, environmental exposure core, integrated health sciences facility core, pilot project program, and career development program, as well as a community stakeholders' advisory committee (CSTAC), representing organizations and communities in the Center's focal region (Figure 1). Each of the Center's subunits had two or three faculty directors, who collectively comprised the Center's core leadership group (CLG). This leadership group was the Center's centralized governing body and they met monthly to discuss research needs and funding opportunities, activities such as research seminars, and other business. The CEC, with counsel from the CSTAC, was primarily responsible for guiding the Center's CER activities. Some of the Center's direct costs were allocated directly to the CEC and were therefore subject to CEC and CSTAC decision making. Other decisions about CER were made by the core leadership group, which included the CEC director. To solicit their feedback and input on Center activities, the 15 CSTAC members were convened twice annually by the staff of the CEC. Logistics and agendas for these meetings were set in conversation with two CSTAC co-chairs who were elected by the CSTAC. Although elections took place once per year, co-chairs were typically re-elected and served for several years. When relevant, CSTAC members also formed research partnerships with the Center's researchers. They were compensated for their time with annual stipends and received additional compensation for their participation in CER projects.

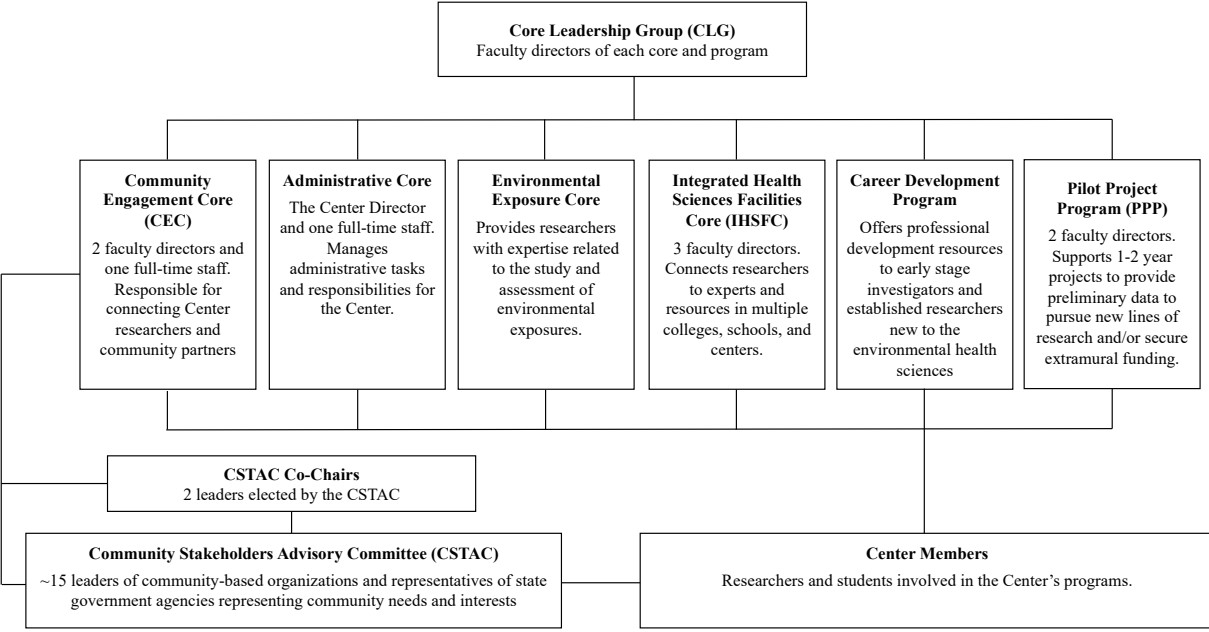

**Figure 1.** Organizational structure of the Center.

The Center's CER efforts in its first funding cycle were designed to orient the Center to the problems and interests identified by those communities at greatest risk from environmental contamination, as well as introducing environmental health sciences (EHS) researchers to the value of CER. The thrust of the Center's CER efforts centered around promoting new CER projects through the Center's pilot project program (PPP). To incentivize researchers to try CER, the Center introduced a policy that all proposals to the PPP

must include a CER plan. The requirements for this plan required that, at a minimum, researchers identify a community partner and define a strategy for working with that partner to disseminate the study's findings to relevant non-academic audiences. Additionally, a community-responsive "Type II" funding mechanism was developed by the CEC and PPP that would provide USD 30,000/year for two years, compared with the USD 30,000/year for one year included in the PPP's "Type I" mechanism. At the time of the research, the PPP required that a portion of this funding be allocated for community partners, but the amount was not specified. At the time of this writing in 2023, a minimum of USD 5000 per year is required to be allocated to the community partner on a Type II project. Projects granted Type II funding were held to much higher CER standards during the review process. These projects would ideally be initiated by communities affected by the issue under study or else address a priority issue identified by the CSTAC.

### 2.2. Procedure

Each of the authors had a role with the Center's community engagement core (CEC) during the Center's first funding cycle. Our roles in the research can be described as participant–conceptualizers (Bennett 1966) through our work with the Center; we engaged in cycles of research, action, and reflection about the goals and purposes of the Center's CER. As staff of the Center, we were privy to discussions, actions, and interactions between leadership that occurred away from the gaze of outsiders, and which laid the foundation upon which other organizational activities rested (Barley 2019). Barley (2019) suggests that such insider perspectives are crucial to understanding how organizations respond to their environments.

Although the Center was required to support CER, we expected that the various people involved in Center activities might have different understandings of CER. To learn more about these various perspectives, the first author conducted semi-structured interviews in 2017, with a convenience sample of Center leaders (n = 3), researchers involved with the Center (n = 2), and CSTAC members (n = 3) (total n = 8). The two Center researchers interviewed were selected because they were, at the time, conducting CER projects with funding and support from the Center and had been closely involved with the CEC. The interview protocol was tailored to each participant's experience with the Center, but all participants were asked to reflect on the Center's CER efforts, the influence of CER programming on their own work, and possible future directions for the Center's CER activities. Due to the relatively small size of the Center, non-gendered pronouns are used to provide some protection of participants' identities.

Audio-recorded interviews were transcribed verbatim by a third party and edited for accuracy. Analysis of these transcripts was informed by our participant observations and goals of improving the Center's CER processes and thus focused specifically on participants' perceptions of CER and the opportunities and challenges of the Center's CER mandate. Transcripts were first coded inductively, yielding themes relating to participants' experience interpreting and enacting requirements for CER, the role of CER in the academic research enterprise, the ideological and logistical obstacles to CER, and the Center's successes and areas for improvement in advancing CER. Perspectives from the Center's community stakeholders' advisory committee (CSTAC) members were particularly important, as the structure of CER has often privileged the needs and interests of academic partners. We therefore held additional focus groups at the CSTAC meeting in the spring of 2018, designed to facilitate reflection on the validity of the preliminary findings and generate new perspectives from CSTAC and other affiliates who had not been interviewed. Following a brief presentation of the study's preliminary findings, attendees, who included 14 CSTAC members, three guest community representatives, and four researchers who were conducting CER projects with funding from the Center (focus group n = 21), were split into two groups and led through an unstructured discussion. Audio recordings of these conversations were transcribed using Otter.ai and corrected by the research team.

Inductive analysis of the focus group data led to the identification of new codes, particularly related to interorganizational collaboration and the challenges of aligning goals, processes, and resources. As themes began to coalesce around organizational structures and processes, deductive codes were introduced that drew from theories of organizational persistence and change. To understand the individual and agentic processes underlying these organizational dynamics, closer reading of key quotations was employed to capture how, through language, participants expressed certain values and priorities (Fairclough 2015).

## 3. Results

### 3.1. Perceptions of Community-Engaged Research

Participants' understandings of CER revealed diverse perspectives informed by personal experience, professional training, and organizational affiliation. Academic participants had perspectives that typically aligned more closely with conventional researcher-driven approaches, while community representatives' interpretations often emphasized a core value of democratic knowledge production. However, "academic" and "non-academic" participants also exhibited very diverse viewpoints; representatives of organizations that directly served communities, for instance, had different expectations for CER than representatives of state agencies, and academics from the social sciences articulated ideas that differed considerably from those of more traditional "bench" scientists. Despite these differences, participants' understandings also revealed several areas of synergy, including the importance of interdisciplinarity and structures that supported more pragmatic, issue- and impact-focused research. In the following sections, we describe several themes that emerged from their perspectives.

#### 3.1.1. Public Scholarship

Most academic researchers involved with the Center reported having little-to-no experience with CER. They joked about their discomfort dealing with the public, alluding to stereotypes of scientists as socially uncomfortable and reclusive. A leader of the Center who identified as a "basic" scientist, for instance, remarked that "basic science is not familiar with the kind of outreach. They think of stakeholders as scary and, you know [...] typically scientists are not outgoing or into community engagement. They want to go pipette something in the room by themself."

When discussing CER, these scientists often described strategies for public scholarship—including such strategies as outreach and dissemination of research—rather than the co-development of research projects with community partners (see McCloskey et al. 2011). Although the NIEHS has embraced engaging the community to help develop research questions, design a study, and collect data, biomedical and health sciences research have typically focused on putting scientific research into the hands of communities and practitioners (i.e., implementation and dissemination) once a study is complete. Through expert panels and other programs, the Center aimed to share research findings with the surrounding community and hired a dedicated communications staff member to share information with the public through social media and other platforms. For several of the academics we spoke with, CER aligned with the university's role as a public institution with their academic service responsibilities. Efforts to gain visibility and positive regard from the public were seen as consistent with the university's responsibility as a land grant institution. Although less common, CER was sometimes implied to be merely "public relations" that had no substantive effect on the research enterprise. Indeed, one researcher described CER as "feel-good stuff," suggesting that the purpose of CER was merely to gain the sympathy and interest of the public.

Some academics also described *education* more than engagement. For instance, several discussed the challenge of using scientific findings to shift people's health-related behaviors, highlighting the possibilities for CER to close this gap. "There's a whole . . . social group blog [...] kind of thing about mothers who smoke [cannabis] during pregnancy because they see it as natural," one researcher explained. "We have to inform them. Okay, folks,

this is what cannabinoids do to the developing brain." They emphasized the challenge of using evidence to change people's minds when they hold a view for which "there is no factual basis ... whatsoever." As one participant explained, "Well, we're in a war on truth. I think community engagement helps folks who are not scientists understand the importance of science. So I think that's really important, actually. Because people are going to vote." These two examples elevate the importance of using CER to correct the public's inaccurate perceptions of health and social matters.

These scientists could be seen as developing their understandings of CER through participation in the Center's CER activities and interacting with the CEC. One scientist remarked, for instance:

> It was a new concept when I first started in 2015 and I was asked 'how are you going to engage the community?' I was like 'research is a silo, community is another silo, we can't talk to each other really.' And then the more I thought about it and the more I spoke to [the CEC], the more it made sense.

This scientist's understanding of academia and their identity as an expert and scientist, in other words, did not include CER, but developed through interactions with CEC staff.

### 3.1.2. Democratizing Knowledge Production

The CEC leadership and staff, who were primarily social scientists with training and experience in CER approaches, drew from established traditions of scholarship and practice to inform their work with the Center. The CEC director, for instance, had been working in the field of community organizing for 20 years before starting work at the Center and employed approaches to participation that were informed by Paulo Freire and other scholar activists who were performing popular education work in the 1980s. Their approach to CER, in other words, reflected a more established set of norms around what CER can and should accomplish. Their previous work had provided them with language, tools, and frameworks for building local power to address social injustices and democratizing who participates in the production of knowledge.

CSTAC members brought diverse perspectives to their work with the Center, with some representing state agencies performing regulatory and policy operations and others working closely with communities to advocate for environmental health and justice. Community organizations were often concerned with protecting communities that had been "studied to death" and gaining access to data that they needed. As one such CSTAC member stated, "communities themselves intrinsically know what the problems are in their own communities, but they don't have any data to back it up." Several advocated for direct involvement of the residents experiencing environmental harms in the research process. They described CER as an opportunity to shift whose priorities were represented in research, recognize the harmful conditions that communities were experiencing, and lend support to efforts communities were already making to protect themselves from environmental harms. These participants underscored the necessity of respecting and valuing the knowledge of affected communities. As one community participant explained:

> when you connect to communities [...] come with a mindset that they're also experts. [...] as researchers, you're gonna go out there and take advantage of that knowledge at the local level because they know what the problems are. They experienced it because they are human test subjects of their own community exposed to all the elements that are there that are in your research projects.

This statement emphasizes the depth of knowledge that residents of these communities have about their own exposures and health effects.

### 3.1.3. Issue-Focused and Transdisciplinary Research

While some conceptualizations of CER did differ considerably between the academic and non-academic participants that we spoke to, others revealed areas of synergy and opportunity. Researchers and community representatives agreed, for instance, that achieving

social impact through the Center's pilot project program (PPP) would necessitate being more intentional about which projects were funded. Describing the pilot projects as increasingly "ad hoc," one community representative described the need for "a long-term plan" for how to address "three major issues in the next two or four years." This CSTAC member, in other words, felt that the projects funded by the Center were not strategic enough in how they would address community-identified issues and yield concrete changes for communities. Several researchers articulated a parallel sentiment about research evidence, suggesting that smaller research projects needed to be more carefully coordinated if they were to generate a body of usable evidence. As one explained:

> our plan is to publish at least one paper, maybe one or two, based on our data, but it's just one out of how many papers published in a year. So if we find more researchers reporting similar data . . . that makes our finding very strong. So I mean, I don't know your budget, but if you can, you could try funding similar projects together.

These academic and non-academic perspectives converged on a more intentional approach to funding research that focuses on issues of concern to the community and other key stakeholders.

For CSTAC members engaged in advocacy work, this approach could help to build stronger evidence bases for their claims. As one such CSTAC member explained:

> those of us who are in the advocacy realm frequently use academic studies to back up the claims that we're making . . . So we are also probably the people that have the best idea what tactics they're using to poke holes in that research. So let's propose another study to look again at it that and focus on those weaknesses or the ways that they're talking this study down.

A key component of that approach, some participants suggested, was to incorporate multiple disciplinary perspectives. One academic, for example, pointed out that "it's worthwhile to have social scientists be a part of your projects and your centers because they can help bring what you do to the communities, they can help bring community concerns to the scientists." These perspectives elevated a pragmatic approach to research, in which the research was developed to fit the intended application.

*3.2. Organizational Change*

3.2.1. Creating the Current: Adherence to Funding Requirements

The Center's leadership expressed a predictable level of concern for their compliance with the funder's CER requirements. Reflecting on the process of applying to the NIEHS for funding to start the Center, a Center leader explained that "it became very clear from the requests for proposals that NIH put out that community engagement was supposed to be a very large part of the center." However, because they were coming from "the sciences where people don't do [CER] normally," they anticipated that "this very important part of [the] center would probably be underdeveloped." To address this potential weakness in their application, "there were a number of center leadership meetings and brainstorming sessions with all the members of the center leadership group about how [they] could strengthen [their] application and make it stronger in every aspect." In conversations with core leaders, we perceived that CER would not have been a high priority if not for the funding requirements. Indeed, several academic participants indicated that, prior to their involvement with the Center, they had been skeptical of CER. One explained how their thinking had changed over time, admitting that "I think previously I just rolled my eyes at the whole concept." Another remarked, "I always thought 'why is it always the researcher benefitting the community' and I had never seen the opposite flow of benefit." Although they often went on to explain that they were now "on board" with CER, the P30 program's funding requirements appear to have been influential in driving these changes.

Indeed, the CEC staff were aware of the obstacles to promoting CER in the Center and saw the P30 program's funding requirements as an important opportunity to "seed the ground" for CER in the health sciences. As the CEC director remarked,

> We are engaging with a scientific infrastructure or scientific legacy that doesn't have that participation in its core. Interestingly enough, the institutional influence here is actually a part of the catalyst. So, the requirements from NIEHS to have community engagement ... that's just this huge influence. And without that, I think we would've been swimming upstream. But in this case it created this whole current. So our job was to kind of to help ride that wave and help the center move along and have the community groups and the researchers all on the boat together, so to speak.

By describing funding requirements as a catalyst, the CEC director suggested that the P30 program's requirements created the conditions that enabled CER to proceed. At the same time, internal organizational processes were necessary to "ride that wave" and produce sustainable strategies.

As expected, community representatives also expressed interest in adapting their organizational structures to enhance collaboration with universities and other organizations represented by the CSTAC. However, increasing CSTAC members' involvement and capacity would require additional resources and the Center could neither afford to provide training for all CSTAC members nor compensate them as part-time staff members. For one CSTAC member, this was a significant weakness of the Center; referring to funding support for partner organizations' capacity development, they explained that "this has been a consistent refrain from the [CSTAC] members. If you want us to do more than we are currently doing, which I think is at this point, kind of bare minimum, there needs to be some concerted effort to try to develop that piece of it."

### 3.2.2. Riding the Wave: New Organizational Processes

Despite the fertile environment created by the Center's CER requirements, building CER infrastructure nevertheless required significant action and effort by committed individuals. The pilot project program (PPP), for instance, "really built over time;" in the program's first few years, the CEC and CSTAC had relatively little involvement in evaluating proposals and applicants were often unaware of the PPP's requirement that all proposals contained a CER component. In subsequent years, the PPP directors helped to clarify these requirements to applicants and CEC directors met with every single PPP applicant to ensure that CER components in their proposals were well-integrated.

For the directors of the PPP, this process required the development of completely new structures and systems. A PPP director explained how they had to alter the proposal review process from the standard NIH approach because they wanted to incorporate the CEC and stakeholders, including the creation of a modified review form for the proposals that had community engagement as a separate scorable component. "We didn't have any infrastructure set up for running the program," they explained. "So the first two years I just did it, but it took like a month out of my scientific life, which is not good for my productivity." Thus, although the PPP was considered a significant achievement in advancing CER, it resulted from unsustainable efforts on the part of several leaders.

Importantly, however, these efforts did yield important advances in CER practice. In its third year, the PPP's Type II mechanism awarded funding to a project that would utilize the Center's interdisciplinary research infrastructure to test several potential contaminants identified by a community. In contrast to following a solely discipline-driven research agenda, this researcher explained, "the community has their concerns and they come to me." Moreover, by leveraging the Center's research infrastructure to examine the community's air, water, and soil, "the [P30] Center [made] it possible to do this really good, holistic research endeavor." This project illustrated how interdisciplinary research infrastructure and CER could be combined to design research that validated communities'

lived experience and to engage them in the research process, values espoused by both academic and community representatives in the Center.

It is important to note that improvements were also made to the Center's CER strategy throughout the duration of this study, particularly as preparations were made for their renewal application in 2020. For instance, although budget constraints meant that all 15 CSTAC members could not be offered significantly higher stipends, a tiered membership system was developed that would increase the leadership responsibilities and compensation of certain CSTAC members. This tiered system was designed to formalize the CSTAC's roles as co-leaders rather than only advisors and to integrate CSTAC members more intentionally into the Center's other activities. Although still working within the constraints of the P30 program's guidelines, the Center's ongoing efforts to improve CER reflect the type of institutional entrepreneurship that can lead to changes in the broader understandings of CER.

### 3.3. Organizational Persistence

### 3.3.1. Mimetic Processes

One of the factors that prevented the Center from developing and adopting innovative CER practices was the tendency to mimic the CER practices of other academic research centers rather than develop approaches that were tailored to the Center's local context. In their recollections about the Center's inception, core leadership described considerable uncertainty about the types of structures and activities that would be considered legitimate by the NIH. To determine what strategies had been funded in the past, one Center leader investigated the practices employed by other similar centers:

> . . . I looked up other pilot project programs at other universities that had centers. [. . . ] like how big were their pilot projects, how often did they solicit, [et cetera]. I created a little sheet of like best practices and I also knew some of the pilot project program directors, like, they were scientists that I knew [. . . ] And so I could call and ask. And they said this really needs to be a really integral part of your center.

Although the CEC staff represented views from the CSTAC in these early conversations, the practices of these other research centers seemed particularly influential in shaping core leaders' perceptions of which CER strategies were legitimate. For instance, although the CEC recommended including a CSTAC member in the funding council for the Center's PPP, it was important to the Center's other leadership that this idea be corroborated by other funded centers:

> The CEC did say [the funding council was] supposed to be set up that way, but so did other center directors on campus. They said you want to have a community stakeholder [on the funding council]. That does well when you're being reviewed.

The opinions of other center directors, in other words, were valued highly as a reference for what would likely be funded by the NIH. Concern for obtaining funding led core leadership to reproduce peers' strategies rather than focusing on input from their own CSTAC and CEC. By looking to their own academic peers for advice, core leadership ran the risk of reproducing existing practices rather than potentially creating more locally appropriate and community-driven approaches.

### 3.3.2. De-Coupling

A second mechanism that hindered the development of new CER approaches was the presence of organizational structures separating community engagement from the Center's other activities. As discussed above, the pressure created by the NIEHS's funding requirements was very influential in driving the creation of the Center's CER practices and organizational structures. Core leaders voiced strong support for CER, and CSTAC members commended the Center for "pushing the envelope" in the integration of CER into its programs. Nevertheless, the CSTAC had very few opportunities to exercise control over the Center's priorities and mainly interacted with the Center via the CEC. This sug-

gested that, in order to reconcile the inconsistencies between CER and standard academic research activities, CER was "de-coupled" (Meyer and Rowan 1977) from the Center's other activities.

The CSTAC played three primary roles in the PPP: annual research needs and opportunity assessments, CER research project partnerships with Center members, and one of the CSTAC co-chair's participation on the PPP's funding council. It is important to note that this co-chair was himself a scientist working for a state agency, but who had earlier experience in the non-profit sector. The research needs and opportunity assessment meeting, which was hosted by the CEC, provided an annual forum in which CSTAC members could propose specific research needs and discuss them with the researchers in attendance. Where potential synergies were identified between researchers and CSTAC members, the CEC would work to broker CER partnerships and help develop proposals to the PPP. The meeting also yielded a summative list of community priorities that was used to guide funding decisions made by the Center's PPP. During the proposal review process, the CEC director and one member of the CSTAC were responsible for scoring proposals according to the relevance and appropriateness of their CER plan. This structure gave the CEC and CSTAC some say over which projects were funded by the PPP and taught other members of the funding council about CER.

Although the integration of CER into the PPP was a novel strategy, the structure of the CSTAC's involvement in the PPP limited their ability to exercise control over the Center's research priorities. Despite being represented on the funding council, the one CSTAC member and the CEC director comprised only a fraction of the council's members and thus had relatively little influence over what was funded. Core leaders explained that "the funding council discusses programmatic priorities of the center, what are our needs? Do we need more zebra fish work, do we need more air pollution work?" at which point they would examine "this nice priority letter that we get from our [CSTAC] that says 'this is what we're interested in' and . . . look at that and . . . say, 'hmm, how many of these applications like meet these needs?' And . . . try to do that." Applicants to the PPP, in other words, were encouraged to propose research that addressed one of the CSTAC's research priority areas, but the funding council ultimately based funding decisions on several competing priorities and could not necessarily prioritize CER.

The CSTAC also had limited opportunities to affect the Center's overall operations. Resource allocation and organizational policies were primarily determined by the Center's academic core leadership group, in which the CSTAC's perspectives were represented via the CEC. CSTAC meetings, which took place twice annually, provided limited opportunities for interaction between the CSTAC, core leadership, and Center members. Most CSTAC members worked a few hours away from the university, so attending meetings either introduced logistical challenges for the CSTAC or for the academics. Attendance by other Center members was also voluntary. As a result, relatively few Center leaders and members regularly attended CSTAC meetings, giving CSTAC members little opportunity to engage directly with the Center's decision makers outside of the CEC. However, the Center director attended most CSTAC meetings, which did provide an important avenue to influence the Center.

Moreover, while these meetings provided a space in which CSTAC members could have given feedback regarding the governing of the Center, most CSTAC members had very limited knowledge of the Center's overall structure and processes and therefore lacked the requisite information to provide such feedback. One CSTAC member remarked, for instance,

> The question that comes up for me is whether the intention of [. . . ] my membership [. . . ] is advisory for your academics to ground their research—make sure it's truly not just nominally community based? Or, was that—was it intended that we actually, like, generate our own ideas that connect us to researchers that make it happen? Is it supposed to be sort of symbiotic or are we advisory?

This question, which was asked in the Center's fourth year of operation, suggests that the CSTAC's role in the Center's activities remained somewhat uncertain. Another expressed "confusion as to where we sit with respect to the Center and the [CEC]," explaining that the Center's subunits and acronyms were difficult to keep track of. They went on to say that, overall, "what our role is hasn't been really clear." These uncertainties were obstacles to CSTAC members' full participation, preventing them from identifying, articulating, and ultimately changing aspects of the Center's operations. The CEC has now developed an extensive job description and on-boarding process to address these concerns.

Finally, the Center's organizational structures, which limited the CSTAC's direct involvement in most activities, left the CEC to represent the views of the CSTAC to the rest of the Center. As only one member of the core leadership group, the CEC sometimes had trouble introducing new policies to the rest of the group. In one meeting, for instance, the CEC suggested that a one-time or annual CER training be required for all Center members in order to build a shared language and set of understandings about CER. Although a few other members of the core leadership group expressed some support for the idea, most expressed concern that the Center's membership would decline if such a training were required and the idea was quickly dismissed. These tensions indicated an ongoing conflict between the dominant academic paradigm and the CER paradigm advanced by the CEC and CSTAC. Because the academic paradigm dominated the core leadership group, opportunities to discuss and resolve these tensions were lost. Based on this experience, training in CER is now required for all Center pilot program grantees.

## 4. Discussion

Community-engaged research (CER) and other forms of community–university collaboration have gained wide acclaim for their benefits to public health and research more broadly, yet studies indicate that their implementation presents ongoing challenges. Despite funding support and incentives from federal funders, descriptive studies of community-engaged research centers point to overworked CER staff, inadequate funding, failure to integrate community stakeholders into research activities, and researchers who do not see the value of CER (Freeman et al. 2014; Holzer and Kass 2015; Wilkins et al. 2013). What these studies do not reveal, however, are the ways that organizational environments, internal structures, and individual agency interact to shape these outcomes.

This study employed perspectives from organizational fields (Fligstein and McAdam 2012) and institutionalism (DiMaggio and Powell 1983; Meyer and Rowan 1977) to examine the organizational dynamics of a federally funded health sciences research center that was required by the funder to incorporate CER into its research activities. By employing an organizational lens, we conceptualized the challenge of CER as an organizational one, resulting not from unmotivated individuals but from complex interactions between community-engaged research centers and their environments. This study shed light on the mechanisms by which academic funding environments, professional and academic training in CER, and organizational structures give rise to the challenges and opportunities of CER in academic health sciences research centers.

This article conceptualized CER in the health and biomedical sciences as an emerging organizational field in which actors from academic institutions and non-academic organizations are increasingly taking one another into account in their actions (Fligstein and McAdam 2012). We asked why, if these organizations are receiving significant funding and support to collaborate, is implementing sustainable CER practices so difficult? The results illustrated how organizational spaces formed for the purpose of interorganizational collaboration, such as community-engaged research centers, are influenced both by the worldviews of each participant and by the logics being constructed for the emerging field.

The findings illuminated how actors in these collaborative spaces drew from their respective organizational and personal worldviews to construct expectations for the emerging field of academic CER, yet also responded to forces, such as CER requirements from funders, thus shaping the collaborative organization itself. Despite a shared commitment



to CER, for instance, some of the Center's leaders responded to CER requirements by mimicking the practices of their academic peers, while others drew from scholarship and practice in public education and community organizing. Non-academic partners' expectations, meanwhile, were informed by their work directly with communities, advocating for health-protective policies or as staff members of state agencies. Respondents' perspectives therefore revealed a range of ideas for how CER should be designed. These perspectives alluded to points of tension between the Center's members, such as fundamentally different views on the validity of community members' experiential knowledge.

Participants' relative power, in turn, determined how negotiations about the "rules of the game" for CER proceeded. Because the P30 program administered funds to academic applicants, the Center was designed and governed primarily by academics, not by community partners. Community partners were engaged, at least initially, in an advisory capacity and were structurally integrated into the Center as an extension of the CEC. This structure—while commonplace in academic CER—hindered community partners' opportunities to engage directly with the Center's leadership and academic members. This separation, when viewed using Meyer and Rowan's (1977) concept of de-coupling, enabled the Center to conform to funding requirements without reducing the Center's ability to generate academic products and outcomes. In other words, core leaders maintained organizational stability by minimizing the opportunity for tensions between academic logics and the, at times, contradictory logics of CER. These findings suggested that, when faced with conflicting institutional logics, the dominant actors in a field may unwittingly hinder creative tension (London et al. 2018) between actors, preventing their disparate views from being effectively negotiated and new understandings reached. Failure to integrate community partners as co-leaders thus undermined the perceived benefits of interorganizational collaboration.

Nevertheless, the findings also highlighted important areas of alignment in actors' views and opportunity in organizational processes for CER. Multiple respondents—both academics and community representatives—emphasized the value of transdisciplinary and team science, coordination of project funding to build bodies of evidence around certain topics and issues, and greater overall intentionality in how project funding was aligned with community priorities. These perspectives aligned with the design of the P30 program, which was intended to provide interdisciplinary research infrastructure to support collaboration and CER. Related to the issues of power and prevention of creative tension, the results also highlighted the promise of "forced" interaction between actors with divergent perspectives. The Center's novel requirement that all pilot projects have a CER plan necessitated that scientists seeking funding needed to interact with the CEC and also ultimately a community partner. These interactions introduced academics inexperienced in CER to the "best practices" for the field, exposing them to the emerging norms for the CER field and expanding their view beyond the norms of their own scientific dogma. These processes also highlighted the critical role of institutional entrepreneurs (Barley 2019) in creating new CER processes and the importance of relationships and social skills in that effort.

These findings have important implications for organizational research and the study of CER. First, viewing CER as an emerging organizational field has several advantages for scholarship and practice. Conceptualizing CER as an emerging field at an institutional and interinstitutional scale widens the view of CER beyond the emphasis on personal values and interpersonal interactions that has dominated existing CER scholarship. Academic CER is as much an interorganizational collaboration as an interpersonal one and community stakeholders' roles include representing the worldviews and interests of their professions and organizations. Their interests, in this case, reflect their organizational contexts and capacities as much as their own personally held values (London et al. 2020). Additionally, characterizing the field of CER in the health and biomedical sciences as emerging can illuminate some of the challenging dynamics that have been observed in empirical studies, as well as guiding future scholarship and practice related to CER. The importance of

developing shared meanings and understandings of CER, for instance, is a key takeaway for practitioners of CER in research centers.

The findings also pointed to several opportunities for research centers to better align their structures with research needs and opportunities in their communities. The first occurred in the interpretation of external messages about what constitutes "legitimate" CER practice. Here, decisions could be made internally about how much to value the perspectives of other research centers versus insights from community stakeholders. While the former could yield concrete insights into the practices and structures that have been funded in the past, the latter accounted for non-academic contexts that shape the issue under study, the research process itself, and the possibilities for the translation of research into public health practice and policy (London et al. 2020).

Internally, structures can determine the extent to which organizational decision making results from synergies between diverse viewpoints. The process of discussing and navigating differences of opinion should be seen as healthy (Bolman and Deal 2017), yielding new knowledge that advances the Center's goals. Research centers should be cognizant of structures that give certain subgroups control over resources, create barriers to participation in decision-making processes, and inequitably distribute important information (Gaventa 1980). However unintentional these effects may be, they can serve to undermine the power of community representatives. One way to facilitate increased and improved interaction between community stakeholders and Center staff is through the development of internal structures that integrate community stakeholders into the range of Center activities. Co-leadership and interaction between academics and community stakeholders can break down barriers between them, creating opportunities for constructive dialogue and co-learning that yields creative new ideas. When limited by constraints on stakeholder compensation and time, research centers can also place a greater emphasis on accountability structures. By, for instance, giving community members greater control over the allocation of funds (e.g., Kegler et al. 2016), research centers can operationalize their commitment to this priority. The importance of social and institutional power in shaping these processes, particularly when working with vulnerable populations, cannot be overstated (Roura 2021).

This exploratory study has generated insights that may guide future scholarly and practitioner efforts. Certain limitations, however, should be taken into consideration. First, this study utilized only a convenience sample of Center leaders, associated researchers, and community stakeholders' advisory committee (CSTAC) members. These perspectives, while providing insights into several of the Center's key CER activities, were neither fully representative nor comprehensive. Future studies should include all leadership and CSTAC members in order to more accurately capture their diverse perspectives. Second, as a single case study, generalizability to other Centers was unclear. To enhance generalizability, efforts were made to provide detail and description to aid in assessing relevance to new settings (Creswell 2013). Additionally, it was expected that features such as resource dependency and basic CER structures would be similar between research centers. Third, the study was performed in the first cycle of the Center that is now finishing its second cycle and therefore did not capture the social learning that has taken place (in part due to these important critiques) since that time.

To conclude, we wish to emphasize the potential ripple effects of organizational CER efforts. There is great need for structures that support the training of transdisciplinary and engaged scholars, timely and community-responsive research, and shortened gaps between research innovation and improvements in health (Petteway et al. 2019). With collective understandings of CER still taking form, there remain considerable opportunities to transform the role of academic institutions in society. The "behaviors," or actions taken by research centers, can be expected to influence broader understandings of CER as organizations adjust their behavior based on the behavior of others (Fligstein and McAdam 2012). In this view, each organization has the potential to either reinforce or disrupt collective understandings of a process such as CER and it is hoped that this study can help

other similar centers ride the wave towards CER that promotes rigor, relevance, and reach ([Balazs and Morello-Frosch 2013](#)).

**Author Contributions:** Conceptualization, K.A.H. and J.K.L.; methodology, K.A.H. and J.K.L.; validation, J.K.L. and K.A.; formal analysis, K.A.H.; investigation, K.A.H.; resources, J.K.L.; writing—original draft preparation, K.A.H.; writing—review and editing, K.AH., J.K.L. and K.A.; visualization, K.A.H.; supervision, J.K.L.; project administration, J.K.L. and K.A.; funding acquisition, J.K.L. All authors have read and agreed to the published version of the manuscript.

**Funding:** The UC Davis Environmental Health Sciences Center is funded by NIEHS grant P30 ES023513.

**Institutional Review Board Statement:** This study was declared exempt from IRB by the UC Davis Institutional Review Board because it did not meet the determination of research.

**Informed Consent Statement:** Informed consent was obtained from all subjects involved in the study.

**Data Availability Statement:** Not applicable.

**Acknowledgments:** We are grateful to the leadership, staff, and community stakeholders of the UC Davis Environmental Health Sciences Center for the valuable insights reported in this manuscript, and to the National Institute of Environmental Health Sciences (NIEHS) for funding and supporting community engagement in environmental health sciences research.

**Conflicts of Interest:** The authors declare that they have no conflict of interest in involved in the production of this article. The funders had no role in the design of the study; in the collection, analyses, or interpretation of data; in the writing of the manuscript, or in the decision to publish the results.

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
