# Peer review of "Creating the Current and Riding the Wave: Persistence and Change in Community-Engaged Health Sciences Research"

_socsci, doi:10.3390/socsci12050312_

Round 1
Reviewer 1 Report
Overall, the manuscript is very well written, provides excellent engagement with sources as well as recent scholarship, and is incredibly insightful and an excellent contribution to scholarship and practice of community engaged research. Minor feedback to the authors:
a. Figure 1 references a community advisory board (CAB) but the text in lines 165-166 references a community stakeholder advisory commitee (CSTAC) which is not in figure 1. Same issue in line 221 where acronym for Community Advisory Board is CSTAC and not CAB. Please clarify if these are the same and use consistent terminology.
b. Similarly, line 168 and 174 references a Center's Center Leadership Group, is this referring to the Core Leadership group (CLG) in figure 1? Please clarify if these are the same and use consistent terminology.
c. Lines 176-177 state that the CSTAC was convened twice annually. Please describe broadly who set the agenda for those meetings and if CSTAC members were invited to co-develop the agenda.
d.Line 185-185 state that proposals to PPP needed to include a CER. Please discribe broadly the parameters or required components of those plans.
e.Line 190, type: "but this no specified level"
f.Line 190, does "currently" refer to 2023? Perhaps use evergreen language "In 2023, a minimum..."
Reviewer 2 Report
(1) This is an important topic and worth exploring.
(2) The introduction could be tightened up a little.
(3) The research questioned should be presented earlier in the paper.
(4) The authors should have mentioned their involvement in the project earlier. I am not particularly concerned about conflict of interest but do worry about objectivity.
(5) Can you speak to generalizability? Do you believe the findings would be different in a non-healthcare research study?
(6) You discussed the results in stage 1 of a project. Are there any results that suggest CER is improving over time as research projects go through later stages?
(7) Are there any best practices that your study points to that can help to improve the likelihood of CER succeeding?
